



**Characterizing water solubility of fresh and aged secondary organic**
**aerosol in PM$_{2.5}$ with the stable carbon isotope technique**
Fenghua Wei[1], Xing Peng[1], Liming Cao[1], Mengxue Tang[1], Ning Feng[1], Xiaofeng Huang[1], Lingyan
He[1]
[1]Laboratory of Atmospheric Observation Supersite, School of Environment and Energy, Peking
University Shenzhen Graduate School, Shenzhen 518055, China.
**Correspondence:** Xing Peng (pengxing@pku.edu.cn)



**Abstract:** The investigation of the water-soluble characteristics of secondary organic carbon (SOC) is
essential for a more comprehensive understanding of its climate effects. However, due to the limitations
of the existing source apportionment methods, the water solubility of different types of SOC remains
uncertain. This study analyzed stable carbon isotope and mass spectra signatures of total carbon (TC)
and water-soluble organic carbon (WSOC) in ambient PM$_{2.5}$ samples for one year and established stable
carbon isotope profiles of fresh and aged SOC. Furthermore, the Bayesian stable isotope mixing (BSIM)
model was employed to reveal the water solubility characteristics of fresh and aged SOC in a coastal
megacity of China. WSOC was dominated by secondary sources, with fresh and aged SOC contributing
28.1 % and 45.2 %, respectively. Water-insoluble organic carbon (WIOC) was dominated by primary
sources, to which fresh and aged SOC contributed 23.2 % and 13.4 %. We also found the aging degree
of SOC has considerable impacts on its water solubility due to the much higher water-soluble fraction of
aged SOC (76.5 %) than fresh SOC (54.2 %). Findings of this study may provide a new perspective for
further investigation of the hygroscopicity effects of SOC with different aging degrees on light extinction
and climate change.
**Keywords:** Fresh SOC; Aged SOC; Water solubility; Stable carbon isotope; BSIM model; Mass
spectrometry.
**Graphical abstract:**

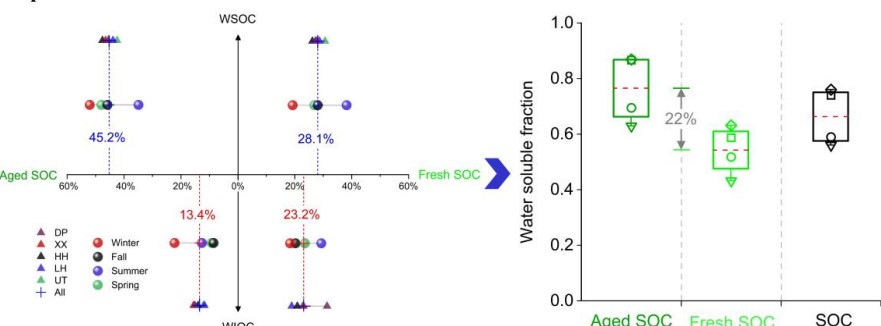



## 1. Introduction

As a major component of particulate matter ($PM_{2.5}$), secondary organic aerosols (SOA) not only contribute to haze formation but also exert a substantial influence on climate dynamics across various spatial scales, from local to global (Kaul et al., 2011; Shrivastava et al., 2017). The water solubility, considered one of the crucial physical properties of SOA, has been extensively studied recently due to its significant effects on the physicochemical processes in the atmosphere. The water solubility of SOA varied with its aging degrees (Kirillova et al., 2013), while both the water solubility and aging degree of organic aerosols contribute to the hygroscopicity noticeably, which affects the light extinction eventually (Han et al., 2022; Liu et al., 2022). Hence, exploring the water solubility characteristics of SOA with different aging degrees can help elucidate the more detailed extinction mechanism of SOA. In addition, recent studies have also shown that the formation of secondary particulates is one of the main processes determining the amount of CCN in remote oceanic regions (Liu and Matsui 2022). Therefore, investigating the water solubility of SOA with different aging degrees is also meaningful for further exploring its indirect climate effects.

Investigating the contributions of SOA with different aging degrees to both organic matter (OM) and water-soluble organic matter (WSOM) is imperative for determining their quantified water solubility. However, due to the constraints of reliable methods, only a limited number of studies have examined the water solubility of SOA using mass spectrometry techniques. Qiu et al. (2019) conducted source apportionment of OM in $PM_1$ and WSOM in $PM_{2.5}$ based on online and offline AMS-PMF methods respectively (Qiu et al., 2019). This approach faces challenges not only related to the inherent errors of online versus offline methods but also discrepancies in the measured particle sizes of OM and WSOM. Kondo et al. (2007) and Timonen et al. (2013) attempted to apportion water-soluble organic carbon



(WSOC) through a multiple linear regression method based on the mass spectral information of OM,
which still exhibits large indeterminateness (Timonen et al., 2013; Xiao et al., 2011; Kondo et al., 2007).
The carbon isotopic technique offers a promising avenue to overcome the aforementioned limitations,
thereby enabling a more in-depth exploration of the water-soluble characteristics of SOA. Carbon isotope
techniques have garnered widespread attention and are increasingly employed in source apportionment
studies of organic aerosols due to their robust source appointment capabilities. Radioactive carbon
isotopes ($^{14}$C) provide a precise method for quantitatively distinguishing between fossil and non-fossil
organic aerosol sources (Fushimi et al., 2011; Zhang et al., 2014). The stable carbon isotope technique
($^{13}$C), however, can quantitatively assess the contributions of various sources by integrating them into
mass balance models (Yao et al., 2022; Widory et al., 2004). The Bayesian mixing model stands out as
one of the most widely utilized models (Xiao;Xu and Xiao 2023; Tang et al., 2020). The stable carbon
isotope technique can also be combined with other source tracers to further enhance the accuracy of
source apportionment of carbonaceous aerosols (Jiang et al., 2022; Plasencia Sánchez et al., 2023;
Ceburnis et al., 2011; Lim et al., 2022). However, to our knowledge, no study has employed the carbon
isotope technique to estimate the source contribution of both fresh and aged SOA before, owing to the
challenging measurement of the carbon isotope profiles for these two sources.

Previous studies have predominantly concentrated on assessing the water solubility of SOA at inland

urban sites, revealing a strong correlation between SOA water solubility and urban air pollution
emissions as well as relative humidity (Wong;Zhou and Abbatt 2015; Pye et al., 2017; Favez et al., 2008;
Salma et al., 2007; Weber et al., 2007; Miyazaki et al., 2006). Nevertheless, few researchers have noticed
the differences between inland and coastal cities. As dynamic interfaces between urban and marine
environments (Donaldson and George 2012), coastal cities exhibit unique characteristics. Shenzhen is a



typical representative city for coastal air pollution studies with a coastline spanning 260.5 km and a total
sea area of 1145 m$^2$. We measured the stable carbon isotope end-members of fresh and aged secondary
organic carbon (SOC), which enables us to investigate the source contributions of SOC with different
aging degrees to WSOC and their respective water solubility in Shenzhen.

The aim of this study is to investigate the water solubility of SOC in PM$_{2.5}$, emphasizing Shenzhen

as a representative mega-coastal city in China. We analyzed stable carbon isotopes and mass spectra
signatures of total carbon (TC) and WSOC in ambient PM$_{2.5}$ samples that were collected from five
distinct sites in Shenzhen over one year as well as specific emission sources. For the first time, we
employed the Bayesian stable isotope mixing (BSIM) model on localized source profiles to quantify the
contributions of fresh SOC and aged SOC to WSOC and water-insoluble organic carbon (WIOC). These
results would contribute to estimating the water solubility of both fresh and aged SOC, revealing their
direct or indirect implications for climate change.
**2. Material and methods**
**2.1 Ambient PM$_{2.5}$ sampling and chemical analysis**
Shenzhen (N22°27′ ~ N22°52′, E113°46′ ~ E114°37′), one megacity of Pearl River Delta, China, is
bordered by Daya Bay and Dapeng Bay to the east, the Pearl River Estuary and Lingding Sea to the west,
Hong Kong to the south, and Dongguan and Huizhou to the north. As a typical mega-coastal city in China,
Shenzhen's air quality is predominantly affected by the continental air mass from northern Guangdong,
the eastern coastal air mass, and the southern marine air mass (Fig. 1). For a comprehensive exploration
of pollution characteristics in Shenzhen, PM$_{2.5}$ samples were collected from five sites covering the
western to eastern regions of the city. The selected sites are Xixiang (XX, urban site), University Town



(UT, urban site), Longhua (LH, urban site), Honghu (HH, urban site), and Dapeng (DP, background site)
(Fig. 1). Additional details about each sampling site are listed in Table S1.

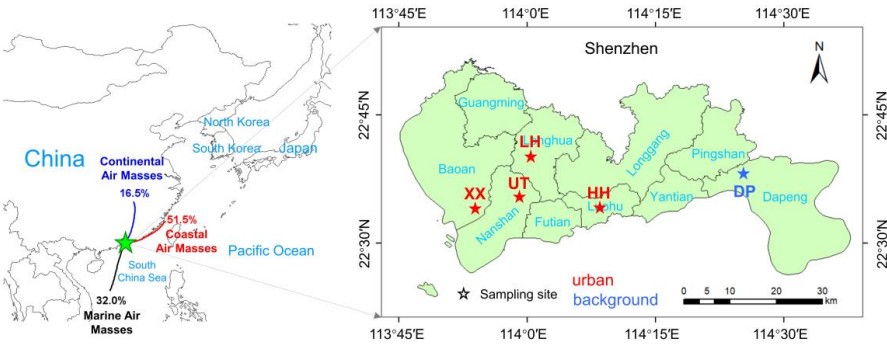


**Figure 1.** Spatial distribution of the five sampling sites in Shenzhen for this study.

In this study, 24-hour $PM_{2.5}$ sampling was conducted every other day in 2019 at the UT site using a

Thermo 2300 atmospheric particulate sampler (Thermo Fisher Scientific Inc., Waltham, Massachusetts,
USA), yielding a total of 160 valid samples. For the remaining four sites, a total of 295 valid $PM_{2.5}$
samples were collected every other day during typical months of the four seasons in 2019 (March, June,
September, and December, Table S2) using a Model TH-16A atmospheric particulate sampler (Tianhong
Corp., Wuhan, China). The organic carbon (OC) and elemental carbon (EC) in $PM_{2.5}$ were analyzed using
an OC/EC analyzer (2001A, Desert Research Institute, Reno, Nevada, USA) following the IMPROVE
A procedure.

For WSOC extraction, the $PM_{2.5}$ sample underwent ultrasonication (20 min × 3 times) in 15 ml

ultrapure water (18.2 MΩ·cm), followed by filtration through a syringe with a 0.45 μm filter head to
eliminate insoluble particles. The extracted $PM_{2.5}$ samples were sequentially analyzed using a long-time-
of-flight aerosol mass spectrometer (L-TOF-AMS, Aerodyne, USA) and an ultrasonic nebulizer
(U5000AT+, Cetac Technologies Inc., USA) to measure elemental ratios, such as O/C, as well as the





mass spectrum signatures of the water-soluble organic fractions, including ion fragments like $CO_2^+$,
$C_4H_9^+$, and $C_2H_4O_2^+$. The concentration of WSOC was determined using a total organic carbon analyzer
(multi N/C 3100, Jena, Germany), and WIOC was calculated as the difference between OC and WSOC.
To investigate the stable carbon isotope signatures of carbonaceous aerosols, we built a stable
isotope spectrometry system by integrating an OC/EC analyzer with a carbon dioxide isotope
spectrometer (QCLAS, Aerodyne). This system reduces the carbon requirement for isotope analysis from
5 µgC to 0.5 µgC and improves the accuracy of spectroscopic measurement methods to 0.2‰~0.3‰.
The stable carbon isotope values of TC and WSOC in ambient $PM_{2.5}$ were measured in this study.
**2.2 Bayesian stable isotope mixing model**
The BSIM model could quantify the contributions of multiple sources to the TC and WSOC based on the
principle of mass conservation of stable isotopes, in which the Markov Chain Monte Carlo (MCMC)
method was employed. The methodology employed in the BSIM model was detailed in works by Parnell
et al. (2013) and Parnell and Inger (2010) (Parnell et al., 2010; Parnell et al., 2013). In brief, the posterior
distribution for the Bayesian neural network (BNN) was calculated utilizing the prior distribution and
likelihood function based on Bayes theorem. Implementation of the BSIM model in this study utilized
the SIMMR package in R software (https://cran.r-project.org/ web/packages/simmr/index.html).
Gelman diagnostic values, ranging from 1 to 1.01, all met the criteria of the posterior prediction test,
indicating robust model performance and reliable results. Additionally, an uncertainty index ($UI_{90}$) was
employed here to further characterize the uncertainty strength of TC and WSOC source apportionments
based on their posterior distribution. This index refers to the difference between the proportional
contributions of the maximum and minimum values in the rapid increase segment divided by 90 with a
90 % cumulative probability ($UI_{90} = (PC_{95}-PC_5)/90$) (Zaryab et al., 2022; Ji et al., 2017).



**2.3 Stable carbon isotope spectrum of PM$_{2.5}$ sources**
The BSIM model requires the input of potential sources for carbonaceous aerosols, along with their local
source-specific stable carbon isotope values (end-members). The PMF model was employed to identify
the TC sources based on PM$_{2.5}$ chemical species concentrations (carbon components, water-soluble
inorganic ions, elements, Text S1), and found traffic sources, secondary transformation sources, and
biomass combustion sources as the major contributors to carbonaceous aerosols in Shenzhen, which are
similar to the previous results in Guangzhou (Huang et al., 2014). Secondary conversion sources could
be further subdivided into fresh SOC for the low oxidation state and aged SOC for the high oxidation
state (Chen et al., 2019; Presto et al., 2009; Mahrt et al., 2022; Shen et al., 2017). Ultimately, traffic
emissions, fresh SOC, aged SOC, and biomass burning (BB) were identified as the four potential sources
of TC and WSOC in Shenzhen in this study.
Recognizing the regional variability in stable carbon isotope fingerprints of PM$_{2.5}$ sources (Yao et
al., 2022), this work obtained representative and locally specific carbon isotope profiles for the four
sources in Shenzhen. For the traffic emissions, we measured the stable carbon isotope values of TC and
WSOC in PM$_{2.5}$ that were collected from the Mount Tanglang tunnel (dominated by diesel vehicles) and
the Jiuweiling tunnel (dominated by petrol vehicles) in Shenzhen. Fresh SOC was simulated through
petrol vehicle bench tests, and the oxygen-carbon ratio (O/C) of PM$_{2.5}$ samples ranges from 0.51 to 0.62,
indicating a low oxidation state of SOA (Ding et al., 2012). The lowest stable carbon isotope values for
TC and WSOC from the simulated sample were chosen as the fresh SOC results. Aged SOC samples
were obtained by collecting ambient PM$_{2.5}$ samples at the National Ambient Air Background Monitoring
Station (Mount Wuzhi site, Hainan, China), primarily influenced by regional pollution transported by
northern continental air masses. These ambient PM$_{2.5}$ samples exhibited a high O/C value of 0.98,



suggesting their highly oxidized state (Zhu et al., 2016). Biomass burning emissions were simulated and
analyzed by burning pine wood in the Laboratory of Biomass Burning Simulation at Peking University
Shenzhen Graduate School (He et al., 2010). Additional details about the sampling process are available
in the Supplementary Information (Text S2). Table 1 summarizes the stable carbon isotope values of the
four sources used in this study. Table S3 compares $\delta^{13}C_{TC}$ source signatures in this study with global
datasets, indicating that the measurement results fall within the range of global datasets. Previous
research identified $C_2H_4O_2^+$ (*m/z 60*) as a reliable marker for biomass burning in Shenzhen, with a feature
value of 1.61±0.68 % (Cao et al., 2018). This prior information was also incorporated into the BSIM
model to estimate the biomass burning source.
**Table 1.** Stable carbon isotope end-members and $f_{60}$ signatures for TC and WSOC sources.

|  | Traffic | | Fresh SOC | | Aged SOC | | BB | |
|---|---|---|---|---|---|---|---|---|
| TC | $\delta^{13}C$/‰ | $f_{60}$/% | $\delta^{13}C$/‰ | $f_{60}$/% | $\delta^{13}C$/‰ | $f_{60}$/% | $\delta^{13}C$/‰ | $f_{60}$/% |
|  | -26.26±0.50 | 0 | -27.31±0.73 | 0 | -25.54±0.28 | 0 | -27.58±0.24 | 1.61±0.68 |
|  | Traffic | | Fresh SOC | | Aged SOC | | BB | |
| WSOC | $\delta^{13}C$/‰ | $f_{60}$/% | $\delta^{13}C$/‰ | $f_{60}$/% | $\delta^{13}C$/‰ | $f_{60}$/% | $\delta^{13}C$/‰ | $f_{60}$/% |
|  | -26.68±0.37 | 0 | -26.18±0.75 | 0 | -24.93±0.39 | 0 | -26.78±0.17 | 1.61±0.68 |

**2.4 Contributions of SOC to WIOC**
Based on the source apportionment results from the BISM model for TC and WSOC, the contributions
of fresh SOC and aged SOC to WIOC were calculated according to the equations (1-2). The uncertainties
(*u*) in concentrations of Fresh SOC $_{(WIOC)}$ and Aged SOC $_{(WIOC)}$ were assessed using the uncertainty
transfer equations (3-4). Fresh SOC and aged SOC uncertainties in both TC (14.9 %, 30.1 %) and WSOC
(24.1 %, 20.9 %) were determined using the BSIM model. Our findings reveal that the calculated
uncertainties of [Fresh SOC $_{(WIOC)}$] and [Aged SOC $_{(WIOC)}$] were 28.3 % and 36.8 %, respectively.

[Fresh SOC $_{(WIOC)}$] = [Fresh SOC $_{(TC)}$]- [Fresh SOC $_{(WSOC)}$]            (1)





$$[\text{Aged SOC}_{(\text{WIOC})}] = [\text{Aged SOC}_{(\text{TC})}] - [\text{Aged SOC}_{(\text{WIOC})}] \quad\quad (2)$$
$$u_{[\text{Fresh SOC}_{(\text{WIOC})}]} = \left( u^2_{[\text{Fresh SOC}_{(\text{TC})}]} + u^2_{[\text{Fresh SOC}_{(\text{WSOC})}]} \right)^{1/2} \quad\quad (3)$$
$$u_{[\text{Aged SOC}_{(\text{WIOC})}]} = \left( u^2_{[\text{Aged SOC}_{(\text{TC})}]} + u^2_{[\text{Aged SOC}_{(\text{WSOC})}]} \right)^{1/2} \quad\quad (4)$$
**3. Results and discussion**
**3.1 Overview of $PM_{2.5}$ and carbonaceous components**
The annual mean concentration of $PM_{2.5}$ in Shenzhen was 24.9 μg/m$^3$ in 2019, with TC being the
predominant component, exhibiting an annual mean concentration of 7.1 μg/m$^3$ (5.8 and 1.3 μg/m$^3$ for
OC and EC, respectively). WSOC accounts for 48 % of OC, presenting an annual mean concentration of
2.8 μg/m$^3$. The mean stable carbon isotope values for TC ($\delta^{13}C_{TC}$) and WSOC ($\delta^{13}C_{WSOC}$) were -26.64 ±
0.79 ‰ and -25.80 ± 0.88 ‰, respectively, which is lower than the results of northern cities in China
(Wu et al., 2020). This can be attributed to the limited impact of coal combustion (which has high $^{13}C$
values) on $PM_{2.5}$ in Shenzhen (Yao et al., 2022; Vodicka et al., 2022).
Seasonal variation revealed that TC, OC, WSOC, and EC exhibited elevated levels in winter and
decreased levels in summer (Fig. 2a). This pattern primarily stems from pollution air masses originating
from continental regions in the fall and winter, and clean air masses from the southern ocean during the
summer months (Fig. S1). The OC to EC ratio, averaging 4.5, was also higher in winter than in summer,
consistent with the Oxygen-to-Carbon (O/C) ratio results for WSOC (Fig. 2a), indicating a large
influence of aged SOC on carbonaceous aerosols in winter. The stable carbon isotope results support this
observation. Figure 2b depicts relatively higher $\delta^{13}C_{TC}$ and $\delta^{13}C_{WSOC}$ values in spring (-26.59‰, -
25.26‰), fall (-26.38‰, -25.44‰), and winter (-26.46‰, -26.27‰). These higher values are attributed
to greater contributions of aged SOC from northern and northeast regional transport processes during



these seasons (Fig. S1). In summer, observed low $\delta^{13}C_{TC}$ and $\delta^{13}C_{WSOC}$ values of -27.29‰ and -26.57‰,
respectively, suggest relatively high contributions of fresh SOC to $PM_{2.5}$. Shenzhen experiences high
temperatures in summer, leading to increased gaseous precursor emissions from terrestrial biogenic
sources, especially C3 plants. Intense solar radiation and high temperature favor photochemical reactions
to generate fresh SOC that depletes $^{13}C$ in particulate matter during summer (Kirillova et al., 2013).

Mass spectra characteristics of $CO_2^+$ (*m/z 44*), $C_4H_9^+$ (*m/z 57*), and $C_2H_4O_2^+$ (*m/z 60*) in WSOC were

measured to represent oxidized organic aerosol (OOA), hydrocarbon-like organic Aerosol (HOA), and
biomass burning organic aerosol (BBOA), respectively. The abundance of these ion fragments, denoted
as $f_{44}$, $f_{57}$, and $f_{60}$, is determined by the ratios of signal intensities at *m/z 44*, *m/z 57*, and *m/z 60* to the sum
of signal intensities from all *m/z* signals in the organic mass spectra. As depicted in Fig. 2c, $f_{44}$ obtained
higher values in spring (0.131) and winter (0.125) compared to summer (0.120) and fall (0.112), further
indicating an elevated oxidation level of OOA during spring and winter. Considering that $f_{60}$ exceeds
0.0030 when biomass burning influences carbonaceous aerosol (Docherty et al., 2008; DeCarlo et al.,
2008), the annual average value of $f_{60}$ was 0.0032, suggesting biomass burning was an important source
of carbon components in Shenzhen. Winter exhibited higher levels of $f_{60}$ (0.0035) compared to other
seasons, suggesting relatively strong impacts of biomass burning on WSOC in winter. Conversely, $f_{57}$
reached its highest level in summer (0.014) and the lowest in winter (0.009), with an annual average
value of 0.011, possibly associated with a notable increase in hydrocarbon organic aerosol emissions
from traffic and biogenic sources during the summer period.





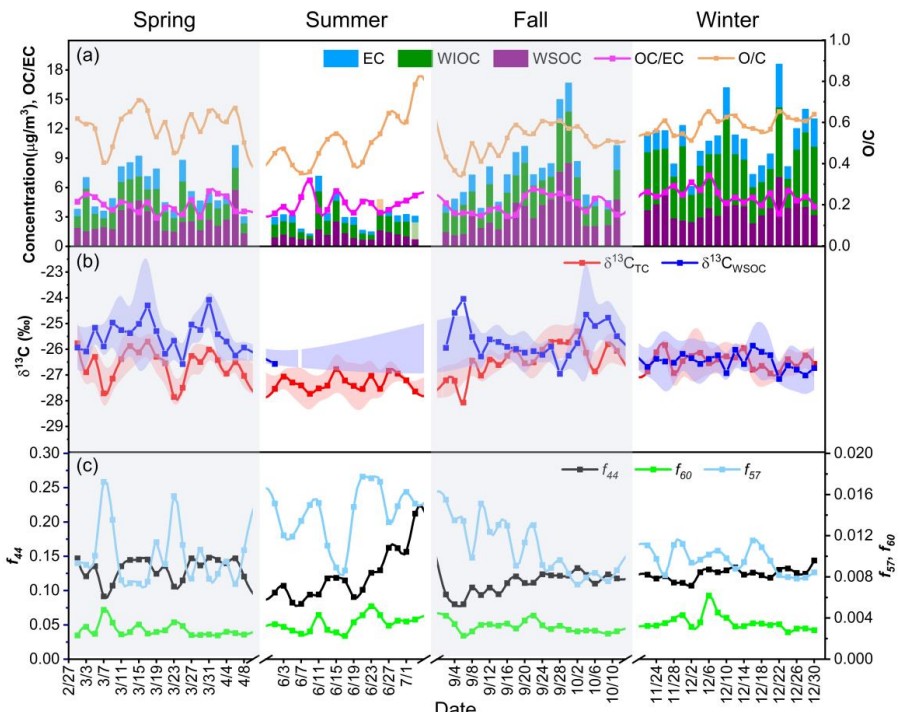

**Figure 2.** Time series of carbonaceous components (a), stable carbon isotope characteristics of TC and

WSOC(b), and mass spectra signatures of WSOC in $PM_{2.5}$ (c) from Shenzhen. Each data was averaged

from five sampling sites. (Note: Summer samples exhibit elevated analytical errors due to low

concentrations, and $\delta^{13}C_{WSOC}$ values are computed from combined summer samples).

Obvious spatial variations in $PM_{2.5}$ mass concentrations across Shenzhen during 2019 were

observed, with XX site registering the highest concentration (29.6 μg/m³), followed by LH (28.0 μg/m³),

HH (23.4 μg/m³), UT (23.1 μg/m³), and DP (20.2 μg/m³). Figure 3 illustrates that TC made more

substantial contributions (28.2 %~32.5 %) to $PM_{2.5}$ at the four urban sites in the central and western

regions of Shenzhen compared to the background site (DP, 25.7 %). This suggests that local pollutant

emissions significantly influence carbonaceous aerosols in Shenzhen's urban areas. The percentage of

WSOC in TC was also higher in urban areas (37.5±3.9 %) compared to the background area (DP, 33.2 %),




reaching the highest value at the LH site (42.9 %). However, the percentage of WIOC in TC displayed
the opposite trend, suggesting carbonaceous aerosols in urban areas of Shenzhen exhibit higher water
solubility than in background areas. Distinct spatial distribution characteristics were also observed in the
stable carbon isotopes of TC and WSOC. The background site exhibits higher $\delta^{13}C_{TC}$ values (-26.33 %)
than the four urban sites (-26.72±0.13 %). This difference may be attributed to the increased contribution
of traffic or fresh SOC sources to carbonaceous aerosols at urban sites and the relatively high contribution
of aged SOC at the background site. Atmospheric aging processes of organics through photochemical
reactions can deplete $^{13}C$ in aged SOC and enrich $^{13}C$ in fresh SOC and other related reactants
simultaneously (Pavuluri and Kawamura 2017). While the close proximity of the $\delta^{13}C_{WSOC}$ values at
urban sites (-25.77±0.04‰) to the background site (DP, -25.96‰) suggests that the WSOC in different
areas of Shenzhen may share a similar origin.



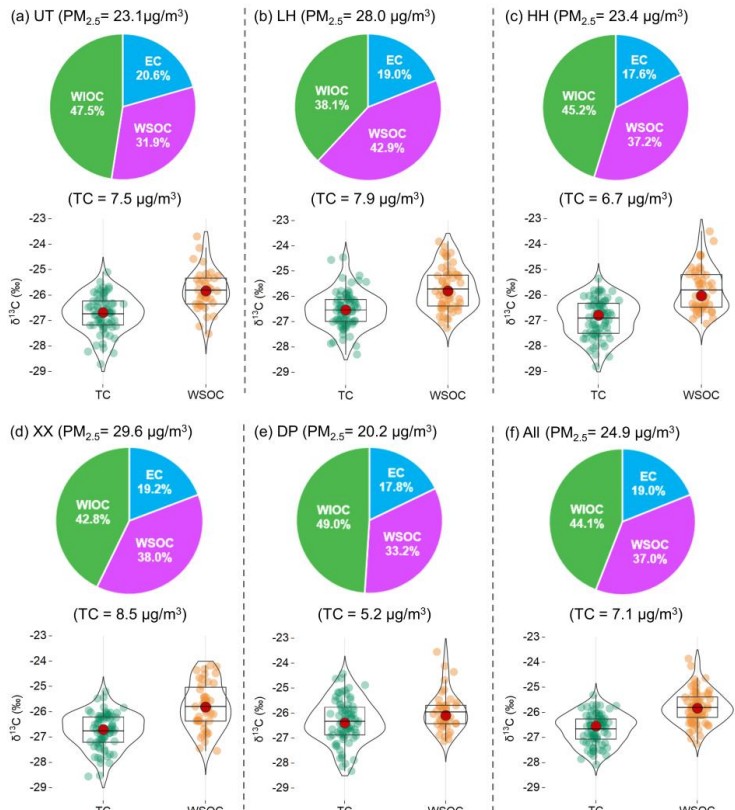

**Figure 3.** Chemical compositions of TC, $\delta^{13}C_{TC}$, and $\delta^{13}C_{WSOC}$ in PM$_{2.5}$ at urban sites (a-d), background

site (e), and average result from all five sites (f). The Violin Box-and-Line Plots on the right display

spatial variations of $\delta^{13}C_{TC}$ and $\delta^{13}C_{WSOC}$ at each site, featuring mean values (black lines) and median

values (red dots)

**3.2 Source apportionment results for TC and WSOC**

The BSIM model assessed the contributions of traffic source, fresh SOC, aged SOC, and biomass burning

(BB) to TC and WSOC, as shown in Fig. 4. On average, SOC (total of fresh and aged SOC) and traffic

emerged as the two major contributors to TC, accounting for 43 % and 40 % respectively, while biomass

burning contributed 17 % to TC. The contribution of aged SOC to TC (23 %) is comparable with fresh



SOC (20 %). Regarding WSOC, SOC was the dominant source, comprising 45 % of aged SOC and 28 %
of fresh SOC, followed by BB (18 %) and Traffic (9 %). The noteworthy contribution of aged SOC to
WSOC suggests a comparatively higher water solubility of aged SOC in Shenzhen.
To evaluate the BSIM model's performance, we employed the PMF model to apportion the sources
of TC and WSOC. The obtained results were subsequently compared with those from the BSIM model,
as depicted in Fig. 4a. Seventeen chemical species of $PM_{2.5}$ were applied as the PMF model input to
estimate source contributions to TC, encompassing carbon components, soluble inorganic ions, and
elements. For the apportionment of WSOC sources, five species including WSOC, WIOC, and three
organic mass spectra were applied as the PMF model input. More details about the PMF model and
results can be found in the Supplementary Information (Text S1, Fig. S2-S4). PMF identified the traffic
as the predominant contributor to TC (55 %), followed by SOC (34 %) and biomass burning (4 %).
Concerning WSOC, aged SOC and fresh SOC were the two major sources as well, accounting for 43 %
and 27 %, respectively. The traffic contribution to TC apportioned by the PMF model is higher than that
of the BSIM model (55 % vs. 40 %), which may be due to the fact that some of the fresh SOC generated
by the conversion of primary vehicle emissions was improperly apportioned to the traffic source in the
PMF model (Li et al., 2022; Zhao et al., 2014). Previous study also showed that SOA contributes more
to carbonaceous aerosols in Shenzhen than the traffic source (Cao et al., 2022). The PMF model results
for WSOC were generally consistent with BSIM model results, with deviations primarily attributed to
the differences in the principles and uncertainties of the two models.
Furthermore, this study examined cumulative frequency distributions to elucidate the inherent
uncertainty in source apportionments of TC and WSOC. As shown in Fig. S5a and b, the proportional
contributions of BB source to both TC and WSOC were quite stable during the research periods due to



its low $UI_{90}$ value (0.02). This may be attributed to the incorporation of mass spectral constraints for the
BB source in the BSIM model used in this study. For TC source apportionment results, the largest $UI_{90}$
value (0.46) was observed for the traffic source, indicating that its contribution to TC exhibited relatively
high uncertainty. In 90 % probability, its contribution ranged from 19.4 % to 60.9 %. The $UI_{90}$ values for
fresh and aged SOC were 0.15 and 0.30, respectively. Regarding WSOC, the calculated $UI_{90}$ value of
traffic, fresh SOC, and aged SOC ranged from 0.18 to 0.24. The $UI_{90}$ values obtained through the BSIM
model remained within reasonable limits, and were smaller than those calculated in previous related
studies (0.23-0.62) (Zaryab et al., 2022; Ji et al., 2017). Consequently, the source contributions of TC
and WSOC estimated by the BSIM model in this study were deemed reasonable.

For seasonal variations, as shown in Fig. 4b, SOC still was the major source of TC and WSOC

during all four seasons, ranging from 38 % ~ 46 % and 71 % ~ 75 % respectively. Significant high
contributions of fresh SOC to TC and WSOC occurred in summer (27 %, 39 %), and relatively higher
contributions of aged SOC to TC and WSOC were observed in winter (26 %, 52 %). It is because
meteorological conditions in winter characterized by inversions and stagnant winds facilitate the
accumulation of air pollutants, and Shenzhen is largely influenced by regional pollution transport in
winter, favoring the formation of aged SOC (Huang et al., 2018). In contrast, favorable meteorological
conditions (e.g. intense and prolonged solar radiation, high temperatures, and relative humidity) in
summer enhanced photochemical reactions to generate fresh SOC. In terms of spatial distributions (Fig.
4c), the contributions of the traffic source to TC were higher at urban sites (38 % to 43 %) compared to
the background site (34 %). This finding aligns with expectations due to increased human activity and
vehicle numbers in urban locations. At the DP site, the contributions of SOC to TC were higher than
those of other sources (47 %), signifying a predominant influence of regionally transported pollutant





emissions on TC at the background site. However, the contributions of SOC and the other two primary
sources at both urban and background sites were all close to each other, indicating the source composition
of WSOC in Shenzhen is less affected by air pollution degree compared to TC.

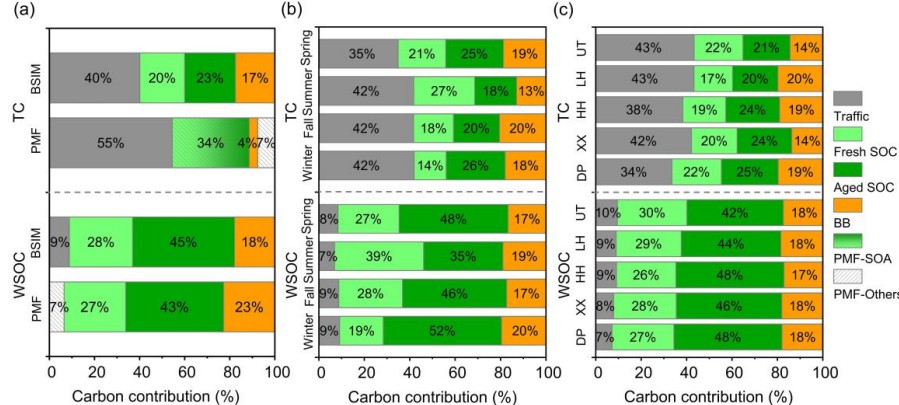

**Figure 4.** (a) Comparison of source apportionment results between BSIM model and PMF model for TC
and WSOC, (b) seasonal and (c) spatial distributions of source apportionment results for TC and WSOC
based on the BSIM model.
**3.3 Water solubility of fresh SOC and aged SOC**
The contributions of fresh SOC and aged SOC to WIOC were the differences between the contributions
of those two SOC sources to TC and WSOC from the BSIM model (Sect. 2.4) in this study. As shown in
Fig. 5a, fresh SOC and aged SOC made contributions of 23.2±4.2 % and 13.4±3.8 % to WIOC,
respectively, implying that primary sources are the dominant contributors to WIOC. Further support for
this finding is evident in the strong correlation between WIOC and EC, as depicted in Fig. 5b and c. A
higher WIOC/EC ratio was observed in winter (2.9) than in other seasons, consistent with the highest
contributions of aged SOC to WIOC in winter (22 %). This observation implies that WIOC in winter is
influenced not only by local primary sources but also by the promotion of secondary pollution.



To investigate deeply the water solubility characteristics of fresh and aged SOC, we then calculate
their water-soluble fraction by comparing their water-soluble portion to the ambient fraction ($[c]_{water-}$
$_{soluble}/([c]_{water-soluble} + [c]_{water-insoluble})$) (Li et al., 2021). As shown in Fig. 5d, the overall water-soluble
fraction of SOC in this study was 66.2 % with a range from 58.9 % to 76.0 %. Fresh SOC exhibited a
much lower water-solubility of 54.2 %, whereas aged SOC displayed a comparatively higher water-
solubility of 76.5 %. The higher water solubility of aged SOC compared to fresh SOC might be due to
the positive correlation between aerosol hygroscopicity and oxidation in the sub-saturated state. The
water-soluble fraction of SOC in this study was close to that reported in other coastal cities (Tokyo (71 %)
and Southeastern United States (60 %)) (Kondo et al., 2007; Verma et al., 2015), while was much higher
than that reported in northern Chinese cities (Beijing (42 % ~ 45 %) and Handan (49 %)) (Li et al., 2021;
Qiu et al., 2019). In addition, the water-soluble fraction of both fresh SOC and aged SOC, as calculated
in this study, was comparable to that reported in Guangzhou (61 % and 86 % for fresh and aged SOC
respectively) (Xiao et al., 2011). This could be attributed to Shenzhen's coastal location, which is
markedly influenced by regional transport from neighboring urban areas and the eastern seaboard air
masses. The high relative humidity facilitates the conversion of aged SOC into WSOC during the
pollution transport process. This result is in accordance with previous findings that air masses influenced
by anthropogenic emissions could promote the formation of high water-soluble SOA under high relative
humidity in urban environments (Miyazaki et al., 2006; Salma et al., 2007; Weber et al., 2007). Given
that the aging process of SOA dissolved in water could enhance the cloud condensation nuclei (CCN)
activity of the particles (Liu and Matsui 2022), high water-soluble aged SOC in Shenzhen might have
significant impacts on the activity of CCN, potentially resulting in more important indirect climate effects.
The water-soluble fraction of SOC (especially aged SOC) in Shenzhen exhibits obvious seasonal



characteristics, with the highest in fall (76.0 %) and the lowest in winter (56.0 %). This phenomenon is
primary related to the robust atmospheric oxidizing capacity during fall in Shenzhen since the
atmospheric oxidants such as OH and $NO_3$ radicals play pivotal roles in driving the secondary generation
of WSOC (Wang et al., 2023). Conversely, during winter, the temperature and relative humidity are at
their lowest levels, and the relatively diminished atmospheric oxidizing capacity also constrains the
secondary generation of WSOC.

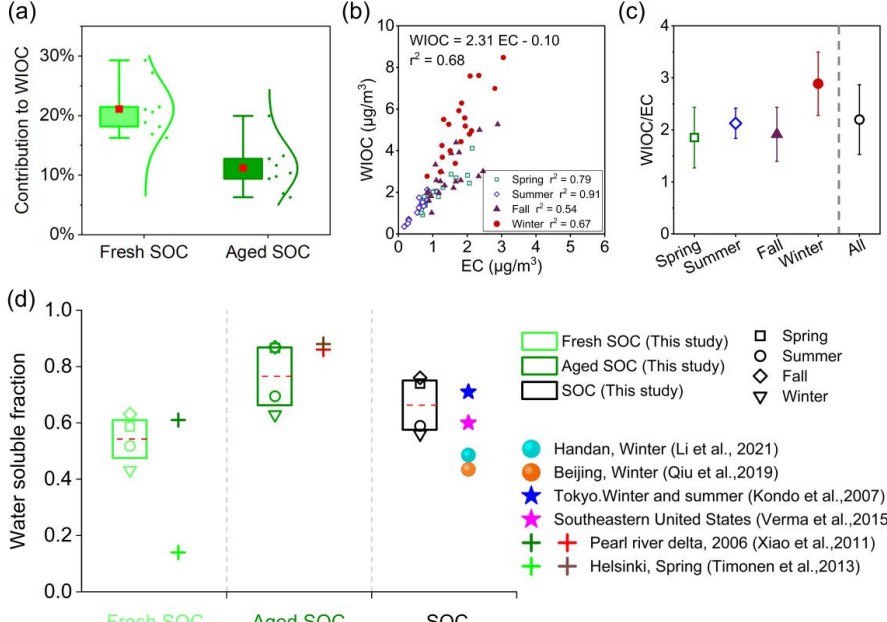


**Figure 5.** (a) Box and whisker plots of fresh and aged SOC contributions to WIOC, the upper and lower
of the box representing the 75th and 25th percentiles, and the red squares featuring mean values, (b)
Scatterplot of WIOC versus EC by season, (c) Seasonal variation of WIOC/EC ratio, (d) Comparison of
the water-soluble fraction of SOC (fresh SOC, aged SOC, SOC) in this study (box and whisker plots)
with those in other related literature (colored markings on the right). The upper and lower of the box
represent the 75th and 25th percentiles and the dashed red lines indicate mean values (Kondo et al., 2007;



Li et al., 2021; Qiu et al., 2019; Timonen et al., 2013; Verma et al., 2015; Xiao et al., 2011).
**4. Summary and implications**
Assessing the impacts of different oxidational SOC on air quality and its water solubility has been
challenging, and this work successfully evaluated the water-soluble fraction of fresh and aged SOC
employing the BSIM model on one-year observational data for stable carbon isotopes and mass spectra
of TC and WSOC in Shenzhen, China. Compared with other methods, e.g. PMF model, EC tracer, and
multiple linear regression analyses, the BSIM model successfully calculated the contributions of fresh
SOC and aged SOC to WSOC and WIOC, owing to prior and localized information about stable carbon
isotopes and mass spectra of $PM_{2.5}$ sources. Therefore, establishing localized carbonaceous aerosol
source profiles for stable carbon isotopes becomes crucial for comprehending the relationship between
the aging degree and water solubility of SOC.

The observed average mass concentration of $PM_{2.5}$ during the sampling period in Shenzhen was

24.9 µg/m³, and WSOC accounts for 48 % of OC. The mean stable carbon isotope values for TC ($\delta^{13}C_{TC}$)
and WSOC ($\delta^{13}C_{WSOC}$) were -26.64 ± 0.79 ‰ and -25.80 ± 0.88 ‰, respectively. WSOC was dominated
by secondary sources while WIOC was dominated by primary sources. The contribution of fresh SOC
and aged SOC to WSOC, WIOC were 28.1 % and 45.2 %, 23.2 % and 13.4 %, respectively. The overall
water-soluble fraction of SOC in this study was 66.2 %, with aged SOC constituting 76.5 % and fresh
SOC 54.2 %. The water-soluble fraction of aged SOC was 22 % higher than fresh SOC, even though
both of them demonstrated remarkable water-soluble characteristics in Shenzhen. This finding highlights
the important role of aged SOC in the water uptake process of particulate matter. Considering the strong
correlation between the water solubility of SOC and its light extinction effect, further exploration of the



extinction effect of SOC with different aging degrees will greatly contribute to a more profound
understanding of the extinction mechanism of SOC. Besides, the water solubility of SOC in coastal cities
was observed to be higher than that in inland cities, suggesting a more pronounced climate effect of SOC
in coastal cities. Therefore, there should be increased emphasis on enhancing the control of SOA
precursors in coastal urban areas to better integrate air pollution and climate change management. This
is particularly crucial given the observed rise in the proportion of SOA in particulate matter in recent
years. Moreover, the results of our study further hinted that the notable water solubility of SOC,
particularly aged SOC, may contribute a lot to the formation of CCN above coastal cities, which is also
helpful to a better understanding of the cloud microphysical processes and the indirect climate effect of
SOC in coastal urban regions.



**Data availability.** Datasets are available by contacting the corresponding author, Xing Peng
(pengxing@pku.edu.cn)

**Author contributions.** PX and HX conceptualized the study. WF, CL, TM and FN retrieved and
constructed the dataset. WF and PX carried out the statistical analysis. WF prepared the first draft of the
manuscript, which was commented on and revised by PX, HL, and HX. All authors reviewed and
approved the final version for publication.

**Competing interests.** The authors declare that they have no conflict of interest.

**Financial support.** This research has been supported by the National Key Research and Development
Program of China (2023YFC3709203) and the Science and Technology Plan of Shenzhen Municipality
(JCYJ20220818100812028).



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
