# Peer review of "Characterizing water solubility of fresh and aged secondary organic"

_EGUsphere, 2024_

## Author Comment (AC1)

We are very grateful for the anonymous reviewer's positive assessments of the manuscript and insightful comments for further improvement. We have revised the manuscript by fully taking the reviewers' suggestions into account. Please find our point-to-point replies below in blue, and the specific changes in the revised manuscript and SI are highlighted here in red.

**Reviewer 1**

Investigating the water solubility of SOA is a highly significant topic, because it has a significant impact on its climatic effects. This work utilized carbon isotopic techniques and mass spectrometry method to evaluate the water solubility of SOA with varying degrees of aging, basing on one-year ambient $PM_{2.5}$ data and established stable carbon isotope profiles of fresh and aged SOA. This work found that SOA has high water solubility, and the solubility of aged SOA is higher than that of fresh SOA. The finding of this work is of great significance for us to deeply understand the climatic effects of SOA. There are certain issues that need to be addressed before considering this work for publication.

1. The source apportionment based on offline data involves water-soluble ions and heavy metal components. The relevant analysis methods should be briefly introduced in the main text and described in detail in the Supplementary Information (SI). Quality control should also be briefly explained.

**Response:**

Thank you very much for your valuable suggestions. The analysis methods of water-soluble ions and heavy metal components have been added in line 105-110 in the revised manuscript, as presented below:

'In addition, the samples collected by Teflon filter in this study were analyzed for water-soluble ions (mainly $SO_4^{2-}$, $NO_3^-$, $NH_4^-$, and $Cl^-$) within $PM_{2.5}$, and the mass concentrations of twenty-three metallic elements (primarily Na, Mg, Al, K, Ca, V, Fe, Ni, Zn, Pb, and Cd) within $PM_{2.5}$ were also determined using an inductively coupled plasma mass spectrometer (ICP-MS, Aurora M90; Bruker, Germany). Relevant quality control information is described in the Supplementary Information (Text S1).'

The detailed quality control methods for $PM_{2.5}$ components have also been included in line 32-41 of the revised Supplementary Information (SI).

'The measuring methods for each component are described in the main text, and the measurement processes were subjected to strict quality control as follows, which are also available in our previous studies (Huang et al., 2019; Yan et al., 2022). The OC/EC analyzer was calibrated using eight standard concentration gradients of sucrose solution prior to each sample analysis of the carbonaceous fractions, with all standard curves achieving an $R^2$ value exceeding 0.999. The charge concentration balance of water-soluble ions ($R^2 = 0.98$, slope = 0.87) confirmed the validity of the measurement results for water-soluble ions. The spiked recoveries for all metal elements ranged between 80 % and 120 % in this study. Furthermore, the background concentration of blank samples and the reproducibility of the measurement results were evaluated during the determination of each component, and all the results met the experimental requirements.'

2. The results of the PMF model should be explained in greater detail, including the explanation of the source profiles identified by PMF and the evaluation of the model results.

**Response:**

Thanks for your suggestion. Detailed explanations of the source profiles identified by PMF and the evaluation of both $PM_{2.5}$ and WSOC model results have been added to Text S1 of SI. The added PMF results and evaluation for $PM_{2.5}$ (line 42-59 in the revised SI) are as follows:

'To find out the optimal solution, factor numbers ranging from 5 to 11 were evaluated using the PMF model. Among them, the nine-factor solution exhibited a notable covariance between vehicle emissions and biomass burning sources, while the eleven-factor solution displayed a dispersed distribution of Pb, Fe and Cd. Subsequently, the ten-factor solution was identified as optimal due to its highly interpretable factor profiles (Fig. S2), with scaled residuals demonstrating a generally symmetrically distribution between −3 and +3. There was a strong correlation between the total mass of the input species and the total mass of all the model-reconstructed factors ($R^2$ = 0.99, slope = 1.04) (Fig. S3), and favorable correlations were also observed between the source contributions and their corresponding source markers ($R^2$ = 0.83 ~ 0.96), suggesting robust performance of PMF model. According to Fig. S2, factor 1 exhibited high percentage explained variation (EV) values for $SO_4^{2-}$ (66 %) and $NH_4^+$ (59 %. In factor 2, not only OM and EC displayed substantial EV values (49 % and 62 %), Zn and Fe also contribute notably. Factor 3 demonstrated the highest EV values for the elements Na and Mg. $Cl^-$ in factor 4 had an EV value of up to 82 %. $NO_3^-$ (67 %) and $NH_4^+$ (25 %) exhibited the highest EV values in factor 5. Factor 6 showed the highest EV values for Pb, Cd and Zn, while factor 7 demonstrated the highest EV values for V and Ni. Factors 8-10 exhibited the highest EV values for Ca (73 %), K (72 %) and Al (76 %), respectively. Consequently, the ten factors were identified as secondary sulfate, vehicle emissions, aged sea salt, coal combustion, secondary nitrate, industrial emissions, ship emissions, construction dust, biomass burning, and fugitive dust, respectively.'

The added PMF results and evaluation for WSOC (line 68-81 in the revised SI) are as follows:

'In the source apportionment of WSOC, the mass concentration and uncertainty matrixes of five species ($CO_2^+$, $C_4H_9^+$, $C_2H_4O_2^+$, WSOC, and WSOO) were put into the PMF model to identify and calculate source contributions to WSOC. Following examination of a range of 2 to 4 factor numbers, a three-factor solution output by the PMF model was determined to be optimal. The scaled residuals exhibited a generally symmetrical distribution between -3 and +3 as well. Moreover, there was also a strong overall correlation between the total factor concentrations reconstructed by the PMF model and the total mass concentrations of the measured species ($R^2$ = 0.99, slope = 0.97) (Fig. S3). According to Fig. S5, factor 1 displayed the highest percentage of EV values for *m/z 44* ($CO_2^+$) and WSOO (73 % and 63 %, respectively), with an oxygen-carbon ratio (O/C) of 1.01, which is highly oxidized and identified as aged SOC source. Factor 2 exhibited EV values of 64% for *m/z 57* ($C_4H_9^+$), 29% for WSOC, 27% for *m/z 44*, and 23% for WSOO. In addition, factor 2 had a lower level of oxidation with an O/C ratio of 0.43, and was therefore identified as fresh SOC source. Factor 3 demonstrated a 100 % EV value for *m/z 60* ($C_2H_4O_2^+$) and a low O/C ratio of 0.36, indicating that factor 3 represented the biomass burning source (BB).'

In addition, Figure S3 was also added to SI to verify the PMF results.

[Figure]

Figure S3. Comparison between the measured total mass of species and the PMF reconstructed total mass of sources of (a) PM$_{2.5}$, (b) WSOC.

3. The uncertainty assessment of the Bayesian model is crucial, it is better to move Figure S5 to the main text.

**Response:**

Thanks for your suggestion. Figure S5 has been moved to the main text (Now Figure 5 in the revised manuscript), as you suggested.

4. To ensure consistency and clarity, it is advisable to arrange the various sources in Figure 4(a) in a uniform order.

**Response:**

Thanks for your suggestion. The TC sources identified by PMF in Figure 4(a) have been arranged in a uniform order in the revised manuscript according to your advice.

[Figure]

Figure 4. (a) Comparison of source apportionment results between BSIM model and PMF model for TC and WSOC, (b) seasonal and (c) spatial distributions of source apportionment results for TC and WSOC based on the BSIM model.

5.  Lines 305-306, the meanings of "$[c]_{water\text{-}soluble}$" and "$[c]_{water\text{-}insoluble}$" should be clearly
explained to avoid any ambiguity.

**Response:**

Thanks for your suggestion. Detailed explanations of the meaning of "$[c]_{water\text{-}soluble}$" and
"$[c]_{water\text{-}insoluble}$" have been added in line 334-337 in the revised manuscript, as presented below:
'we then calculate their water-soluble fraction by comparing their water-soluble portion to the
ambient fraction ($[c]_{water\text{-}soluble}/([c]_{water\text{-}soluble} +[c]_{water\text{-}insoluble}$), where $[c]_{water\text{-}soluble}$ and $[c]_{water\text{-}insoluble}$
are the concentrations of fresh SOC or aged SOC in WSOC and WIOC, respectively) (Li et al.,
2021).'

6.  Figure 5(c) appears redundant as it overlaps with Figure 5(b) in terms of information
presented. To streamline the content, it is advisable to include the slope information within Figure
5(b).

**Response:**

Thanks for your comments. We have followed your suggestions and made corresponding
adjustments to Figure 5 (Now Figure 6 in revised manuscript).

[Figure]

**Figure 6.** (a) Left is the box and whisker plots of fresh and aged SOC contributions to WIOC, the upper
and lower of the box representing the 75th and 25th percentiles, and the red squares featuring mean
values. The dots on the right show the contribution of fresh and aged SOC to WIOC across seasons and
sites, the curve demonstrates its normal distribution. (b) Scatterplot of WIOC versus EC by season, (c)
Comparison of the water-soluble fraction of SOC (fresh SOC, aged SOC, SOC) in this study (box and
whisker plots) with those in other related literature (colored markings on the right). The upper and lower
of the box represent the 75th and 25th percentiles and the dashed red lines indicate mean values.

7.     Line 38, the full name of CCN should be clearly listed at the first mention in the main text.

**Response:**

Thanks for your suggestion. The full name of CCN (cloud condensation nuclei) has been added
in line 38.

---

## Author Comment (AC2)

We are very grateful for the anonymous reviewer's positive assessments of the manuscript and
insightful comments for further improvement. We have revised the manuscript by fully taking the
reviewers' suggestions into account. Please find our point-to-point replies below in blue, and the
specific changes in the revised manuscript and SI are highlighted here in red.

**Reviewer 2**

The manuscript analyzed the water-soluble and water insoluble organic carbon in a coastal megacity
of China. The sources and contributions of WSOC and WIOC to $PM_{2.5}$ were explored with Bayesian
stable isotope mixing model, and the water solubility of fresh and aged SOC in the coastal megacity
of China were revealed. The study is important and meaningful, while there are still some questions
need to be clarified to improve the manuscript.

**Specific comments:**

1.   I am confused about how the BSIM and PMF model used in the source apportionment of the
TC and WSOC. The author should re-organize section 2.3 to make it clear.

As described in section 2.3, The four sources, including traffic, biomass burning, the fresh SOC and
aged SOC were resolved by PMF model. The author also obtained the stable carbon isotope
fingerprints by traffic emission samples collected in tunnels, fresh SOC simulated through petrol
vehicle bench tests, aged SOC samples collected at a background monitoring station and biomass
burning samples simulated through laboratory experiment.

Did the fingerprint above were used as prior information in the BSIM model? What about the $\delta^{13}C/‰$
values of the four sources? How could the author verify the representation of the four source
fingerprints and that they can be properly used in Shenzhen?

**Response:**

We sincerely appreciate your suggestion. The PMF model referred to in section 2.3 of the
manuscript serves to determine the potential source of TC and WSOC for further refined source
apportionment using the BSIM model. We have already re-organized section 2.3 to make it much
clear. At lines 142-147 in the revised manuscript, we have provided a more detailed description of
the role of the PMF model in this study:

'In this study, we firstly employed the PMF model to identify the potential sources of TC and
WSOC (Text S1), with the aim of reducing the uncertainty of the subsequent BSIM model and
verifying the reliability of the BSIM results. The PMF results showed that traffic emissions, SOA,
and biomass burning are the major contributors to carbonaceous aerosols in Shenzhen, which were
similar to the previous results in Guangzhou (Huang et al., 2014).'

In this study, stable carbon isotope fingerprints of traffic emissions, fresh and aged SOC, and
biomass burning sources were obtained, along with mass spectral signatures of biomass burning
source $(f_{60})$, and all these fingerprints were used as prior information for the BSIM model.
Additional clarification on this point has been incorporated at lines 159-160 in the revised
manuscript:

'The measured profiles of the four sources were used as prior information in the BSIM model for the follow-up analyses.'

The specific $\delta^{13}C/‰$ values for the four sources were detailed in Table 1. The four source fingerprints used in this study were all compared with the results from relevant literatures in Table S3 in the Supplementary Information (SI). The comparison demonstrated that the stable carbon isotope values of each source measured in this study closely aligned with literature findings and fell within the range of source spectra reported in other studies, which confirms the reliability of the four source fingerprints employed in this study. Regarding this point, we have also included further elucidation at lines 175-177 in the revised manuscript:

'The stable carbon isotope measurements from the four sources align with the range observed in global datasets, thus affirming the reliability of the four source fingerprints utilized in this study.'

2.    This study firstly employed BSIM model to quantify the contributions of fresh and aged SOC to WSOC and WIOC. The author claimed the consistence of the results from BSIM and PMF model. My question about the method used in this study is what is the advantages of the BSIM model compared with PMF model? Why was the result of BSIM model used for the final analysis?

**Response:**

Thank you for your comment. The most significant advantage of the BSIM model over the PMF model in this study is its capability to simultaneously quantify fresh and aged SOC in TC and WSOC. In the source apportionment of offline $PM_{2.5}$ samples based on the PMF model, the absence of mass spectrometry information makes it impossible to differentiate between fresh and aged SOC in TC. Consequently, only the overall SOC source can be apportioned. Therefore, PMF model could not provide further quantification of the water solubility characteristics of SOC. Hence, the BSIM model results were employed in this study as the final analysis. This point has been clarified in section 2.3 of the revised manuscript (lines 151-156):

'Since the PMF model lacks the mass spectral information of offline $PM_{2.5}$ samples, it fails to distinguish between fresh SOC and aged SOC in TC, making it challenging to investigate the water solubility characteristics of the SOC based on PMF results. BSIM model simultaneously quantified of fresh and aged SOC separately in both TC and WSOC, thereby enabling an estimation of SOC water solubility. This capability is used for the final analysis in this study.'

3.    Line137-138, the SOC was divided into fresh SOC and aged SOC based on the oxidation state, what is the exact values of the average oxidation state of carbon (OSc) or O/C of the two SOC sources?

**Response:**

Thank you for your comment. The O/C ratios ranged from 0.51 - 0.62 for fresh SOC and the mean O/C ratio for aged SOC was 0.98 in this study, both of which were close to the O/C ratios calculated by the PMF model for fresh SOC (0.43) and aged SOC (1.01). We have improved the accuracy and clarity of the descriptions in lines 164-166 and lines 169-170 of the revised manuscript:

lines 164-166: 'The oxygen-carbon ratios (O/C) of fresh SOC samples in this study ranged from 0.51 to 0.62, indicating a low oxidation state (Ding et al., 2012).'

lines 169-170: 'These aged SOC samples exhibited a high O/C value of 0.98, suggesting their highly oxidized state (Zhu et al., 2016).'

4. Line 162, table 1 showed the $\delta^{13}C$/‰ values of the four sources in TC and WSOC, I noticed that the value of $\delta^{13}C$/‰ for different OC showed obvious overlap, for example, the values of fresh SOC in WSOC and TC were lower and higher than that of traffic source, respectively? How did the sources were determined being clearly separated by the values of $\delta^{13}C$ with the existence of the obvious overlap of the $\delta^{13}C$/‰?

**Response:**

Thank you for your comment. Stable carbon isotope values of atmospheric particulate matter from different sources may overlap for a number of reasons. For instance, the values of fresh SOC and traffic emissions are likely to overlap primarily because some fresh SOC could be further generated by traffic emissions, and different components of the same source may possess similar stable carbon isotope values. Besides, particulate matter fractions from different sources may undergo distinct physical and chemical processes in the atmosphere, along with carbon isotope fractionation, all of which could contribute to changes in stable carbon isotope values.

Although the overlap degree of stable carbon isotope values from different sources may affect the accuracy of the Bayesian approach, it's essential to note that the Bayesian approach is a probabilistic model that utilizes both a priori information and a likelihood function to estimate the contribution of sources. This approach enables probabilistic estimation of contributions from different sources. In addition, it can also integrate information from multiple markers and sources, thereby mitigating the effects of overlap and enhancing the robustness of source apportionment analyses.

In this study, the following methods were employed to mitigate the impact of $\delta^{13}C$ values' overlapping between different sources on the BSIM model. First, the number and type of potential sources of TC and WSOC were identified in advance based on the PMF model, avoiding the uncertainty of interference from unrelated sources. Secondly, in addition to stable carbon isotopes, the chemical tracer marker of biomass burning source ($f_{60}$) provided additional information to the BSIM model, which helped to improve not only the apportionment accuracy of biomass burning source, but also the overall accuracy of the BSIM model.

The following content has been added to section 2.3 in the revised manuscript (lines 179-184): 'Although there is some overlap among the $\delta^{13}C$ fingerprints of different sources, the Bayesian approach allows for probabilistic estimation of the contribution of different sources and can also integrate information from multiple markers and sources to mitigate the effects of overlap. In this study, the PMF model was used to reduce the uncertainty of interference from unrelated sources, and the chemical tracer marker of biomass burning source ($f_{60}$) was also integrated to minimize the effect of this overlap.'

5. Line332, the meaning of the dots and lines in Figure 5a should be added.

**Response:**

Thanks for the suggestion. The meaning of the dots and lines has been added in the caption of Figure 5a (Now Figure 6a in revised manuscript). The specific additions are as follows:

[Figure]

**Figure 6.** (a) Left is the box and whisker plots of fresh and aged SOC contributions to WIOC, the upper
and lower of the box representing the 75th and 25th percentiles, and the red squares featuring mean
values. The dots on the right show the contribution of fresh and aged SOC to WIOC across seasons and
sites, the curve demonstrates its normal distribution. (b) Scatterplot of WIOC versus EC by season, (c)
Comparison of the water-soluble fraction of SOC (fresh SOC, aged SOC, SOC) in this study (box and
whisker plots) with those in other related literature (colored markings on the right). The upper and lower
of the box represent the 75th and 25th percentiles and the dashed red lines indicate mean values.

6.    Line 338-339, the reference has been listed in the legend of figure 5, so it doesn't need to be
listed in the caption here.

**Response:**

Thanks for the suggestion. The references have been removed from the caption of Figure 5
(Figure 6 in revised manuscript).

7. The subdivided OOAs were name as fresh SOC and aged SOC in the manuscript, while they were
named MO-OOA and LO-OOA in Figure S4. Please make the names of these items consistency
through the manuscript or add some statement of the difference.

**Response:**

Thank you very much for raising this point. Since fresh SOC and aged SOC, as well as LO-
OOA and MO-OOA, were differentiated by the degree of oxidation of SOC in our study, and the
O/C ratios of fresh SOC and LO-OOA, as well as aged SOC and MO-OOA, were close to each
other, we have adjusted the names of the two to be consistent as you suggested. Specifically, we
have renamed LO-OOA and MO-OOA in Figure S5 and SI as fresh SOC and aged SOC, respectively.

'According to Fig. S5, factor 1 displayed the highest percentage of EV values for *m/z 44* ($CO_2^+$) and

WSOO (73 % and 63 %, respectively), with an oxygen-carbon ratio (O/C) of 1.01, which is highly oxidized and identified as aged SOC source.  Factor 2 exhibited EV values of 64% for *m/z 57*

($C_4H_9^+$), 29% for WSOC, 27% for *m/z 44*, and 23% for WSOO. In addition, factor 2 had a lower level of oxidation with an O/C ratio of 0.43, and was therefore identified as fresh SOC source.'

[Figure]

   **Figure S5.** The source profiles resolved by PMF for WSOC.